# A Structural Equation Modelling Approach to Examine the Mediating Effect of Stress on Diet in Culturally Diverse Women of Childbearing Age

**DOI:** 10.3390/nu16193354

**Published:** 2024-10-02

**Authors:** Karim Khaled, Fotini Tsofliou, Vanora A. Hundley

**Affiliations:** 1Department of Public Health, Faculty of Health, Education, & Life Sciences, Birmingham City University, Birmingham B15 3TN, UK; karimjkh@gmail.com; 2Department of Rehabilitation & Sport Sciences, Faculty of Health & Social Sciences, Bournemouth University, Bournemouth BH8 8AJ, UK; ftsofliou@bournemouth.ac.uk; 3Centre for Wellbeing and Long-Term Health, Faculty of Health & Social Sciences, Bournemouth University, Bournemouth BH8 8AJ, UK; 4Centre for Midwifery and Women’s Health, Faculty of Health & Social Sciences, Bournemouth University, Bournemouth BH8 8AJ, UK

**Keywords:** psychological stress, stress, diet quality, dietary patterns, women, females, childbearing age, reproductive age, structural equation modelling

## Abstract

**Background**: Stress has been shown to be associated with poor nutrition among young women. However, studies around the topic have major limitations in their methodologies and the role of confounding factors within this association remains unclear in the literature. **Objective**: To investigate the associations between stress and dietary quality/patterns in a culturally diverse population of childbearing-aged women. **Methods**: A secondary analysis of data from two studies conducted in UK and Lebanon was performed using Structural Equation Modelling (SEM) to explore the role of country context, socio-economic status, and physical variables. Participants were healthy women of childbearing age (*n* = 493). Variables measured were dietary intake, stress, physical activity, sociodemographic variables, and Body Mass Index. These were included in the SEM analysis to examine the mediating effect of stress. **Results**: The results showed that, among all variables, only country had a significant effect on dietary quality and patterns through the mediatory effect of stress. Participants from Lebanon were found to have higher stress levels compared to participants from the UK, and this contributed to a lower adherence to a Mediterranean diet. **Conclusions**: This study shows that for women of childbearing age, having a good diet quality/pattern is dependent on stress levels and country context.

## 1. Introduction

Evidence indicates that ensuring women are in good physical and mental health for pregnancy will benefit both them and future generations [1,2]. Obesity among women of childbearing age has risen rapidly in the last two decades and has been associated with gestational diabetes, preeclampsia, miscarriage, and a variety of cardiovascular illnesses that place both the mother and the baby in danger [3,4,5,6].

Within this context, recent research in nutritional epidemiology has reported that higher dietary quality (DQ) and healthier dietary patterns (DPs) have been linked to a lower risk of obesity and obesity-related disorders (type-2 diabetes, cardiovascular disease, mental illness) [7]. Western dietary patterns (high in saturated fat, sugar, and refined/processed foods) reflect a lower diet quality that promotes obesity and chronic disease [1]. In contrast, a Mediterranean-style diet (high in fruits, vegetables, and fresh produce), traditionally consumed in the Mediterranean region (e.g., Lebanon and South Europe), has been associated with reduced obesity rates and non-communicable diseases [8]. In epidemiological research, the diet of a population can be descried through either “a priori” or “a posteriori” methods. The “a priori” method involves the calculation of a dietary index or score, which has been found to be inversely associated with disease outcomes and assesses the adherence of a participant’s food intake to that dietary index [9]. A commonly employed “a priori” dietary index is the Mediterranean diet (MD), which has been shown to be protective against cardiovascular and non-communicable diseases (e.g., cancer, diabetes, hypertension) [10]. The MD dietary index was developed by Trichopoulou et al. [10] to evaluate adherence to the Mediterranean diet and involves the analysis of nine food categories: cereals, vegetables, fruits and nuts, meat, dairy products, seafood, alcohol, legumes, and unsaturated fats. Conversely, the “a posteriori” method employs data-reduction techniques, such as factor analysis, to identify the dietary patterns of participants that are derived by assessing and collecting data regarding their food intake [9]. This method identifies dietary patterns by clustering food groups that exhibit high correlations and determines the number of dietary patterns and their components (food groups within each dietary pattern) [11]. The integration of both methods in nutritional research enables a comprehensive assessment of dietary patterns and their associated factors through both data-driven (a posteriori) and hypothesis-driven (a priori) approaches [12].

There is considerable heterogeneity in dietary patterns across the world and this can result in different outcomes in terms of a population’s health [13]. It is vital to understand the factors that influence dietary patterns in different countries to provide a strong evidence-base from which to support international interventions aiming at improving diet quality and public health [13]. This is particularly important in relation to pregnancy and childbirth because both mothers and newborns are affected by dietary choices [14].

Stress is a major driver of poor diet quality especially among young women [15,16]. Research has shown that young females (18–29 years old) who report high levels of perceived psychological stress are more likely to consume fatty foods and less likely to consume fruits and vegetables than non-stressed females [16,17,18,19,20,21,22,23,24,25]. However, the approaches taken by studies that looked at stress and diet have been limited, with most studies assessing specific food groups and their relationship with participants’ stress, rather than evaluating the dietary patterns and quality as a whole-food dietary approach. For example, Barrington et al. [25] assessed fast food intake as the outcome rather than the whole diet pattern. This does not provide a representative picture of real-life dietary intake and habits. Additionally, the study used a single item scale, and this is not deemed accurate in the dietary assessment of fast-food consumption.

Psychological stress has recently been recognised as a major risk factor for poor nutrition and its associated health complications, but other factors such as age, low socioeconomic status, high BMI, and physical inactivity have also been found to affect health [26,27,28,29]. The direct and indirect mechanisms underlying the association between stress and diet remain unclear. This is a result of the unquantified interrelationships and elevated collinearity among psychological factors and other behavioural and lifestyle variables. There is a need to explore how these lifestyle factors interact with stress and their potential influence on diet. A conceptual model, bringing together the various factors, would enable a better understanding of this complex issue. Structural Equation Modelling (SEM) is a leading method for assessing conceptual models by quantifying the interactions among a network of variables. SEM has the feature of assessing all associated routes at the same time, taking into account the function of independent and/or dependent (i.e., mediator) components in outcome formation [30].

Studies with rigorous methodologies and designs from different countries (providing cultural and socio-demographical diversity) are needed to appropriately address the association between stress and diet in women of childbearing age [31]. Most of the evidence to date has come from studies conducted in single countries, where participants have the same culture, traditions, and nutritional habits, which may eventually influence the perceived psychological stress level, dietary intake, and the association between them [22]. Moreover, there are differences in these studies’ methodologies, variables assessed, and tools used for data collection, which make the comparison between single-country studies difficult [32]. Few studies have recruited participants from different countries, and the one study that has drew its samples from the same continent (Europe) with broadly similar food cultures, sociodemographic status, and stress levels [22].

Robust research is needed to enhance our understanding of the relationship between stress and diet in women of childbearing age. Ensuring that the research involves diverse populations from different countries and continents will enable us to investigate the different factors that could affect this relationship. To the best of our knowledge, no previous study has evaluated the association (direct and indirect) between perceived stress and dietary quality and patterns among women of childbearing age from different regions of the world. We utilised Structural Equation Modelling (SEM) to investigate the association between perceived stress and dietary quality and patterns in a culturally diverse population of women of childbearing age from the UK (Europe) and Lebanon (Middle East). By including two countries in different regions, we are able to explore the role of confounding/explanatory factors such as country context, physical variables, and socioeconomic status.

## 2. Materials and Methods

An online survey questionnaire was used to collect data from the United Kingdom (Europe) and Lebanon (Middle East) [33,34]. Written informed consent was obtained from all participants in both countries on the landing page of the surveys. Ethical approvals in the two countries were obtained through the Ethics Committee at Bournemouth University in the UK (protocol code 22344) and the Institutional Review Board at the Lebanese American University in Lebanon (IRB#: LAU.SAS.MB3.27/Nov/2020). The study reporting follows the Strengthening the Reporting of Observational studies in Epidemiology (STROBE) guideline [35].

### 2.1. Study Population

In total, 493 women of childbearing age participated in both studies (*n* = 244 in UK and *n* = 249 in Lebanon). Participants were recruited as a convenience sample of women of childbearing age through classroom visits, flyers and posters, and social networking sites. The enrolment procedure of participants is shown in Figure 1. To participate in the study, participants had to meet the following inclusion criteria:Do not have chronic disease (e.g., cancers, diabetes, cardiovascular diseases, HIV/AIDS, multiple sclerosis, pulmonary diseases, or mental disorders);Do not suffer from food intolerance/allergy;Are not pregnant or breastfeeding;Are not on medications that impact appetite;Have not previously had a bariatric surgery.

### 2.2. Variables

Lifestyle and psychosocial variables were assessed and included dietary intake, psychological stress, adiposity, physical activity, and socio-demographic data (such as income, marital status, age, and ethnicity). Further details can be found elsewhere [33,34]. A brief description of the scales used is given below.

Dietary intake was measured using the European Prospective into Cancer and Nutrition Food Frequency Questionnaire (EPIC FFQ), which has been previously validated in the UK [36] and Lebanon [37]. Dietary quality and patterns were derived through a priori and a posteriori approaches. The a priori approach was based on assessing the adherence to Mediterranean diet [10], and the a posteriori approach was based on performing factor analysis to derive the latent dietary patterns of participants [33]. Both approaches are discussed below. Dietary data collected from the EPIC food frequency questionnaire was analysed using the FETA software where the grams/day of eleven food groups were derived to be used in assessing the diet quality and patterns [38].

Stress was assessed by the Perceived Stress Scale (PSS) [39], which was previously validated in the UK [40] and Lebanon [41]. Using the PSS, participants were categorised into four quartiles: the lowest stress levels with scores ranging from 0 to 14, a middle–low stress level with scores between 14 and 28, a middle–high stress level with scores from 28 to 42, and the highest stress levels with scores ranging from 42 to 56 [39].

The physical activity level of participants was assessed via the International Physical Activity Questionnaire (IPAQ), which has been validated in UK and Lebanon [42,43]. Body Mass Index (BMI) was calculated by dividing weight (kg) over height squared (m^2^) where weight and height were self-reported by participants in the surveys. Information on socio-demographic characteristics (age, marital status, income, ethnicity, and smoking status) was also collected. 

### 2.3. Building the SEM Model

After identifying the research problem in the introduction of this paper as a first step, the hypothetical model was identified in the second step. The model hypothesises that the dietary quality (adherence to Mediterranean diet) and dietary patterns (derived from factor analysis) were deemed as dependent variables, whereas stress was deemed the mediating variable relating the sociodemographic characteristics, BMI, and physical activity (exogenous variables) to the dependent/endogenous variables (dietary quality and patterns). The model was built based on evidence from the literature as follows: the country of participants along with their sociodemographic characteristics, adiposity measure, and physical activity level were expected to be directly associated with stress, and stress was allowed to predict the dietary quality and patterns. The evidence-based justification of the hypothesised model is discussed below.

Previous studies have reported that sociodemographic characteristics are associated with stress. For example, age and ethnicity have been found to be linked with stress where the literature reports that older people tend to have lower stress levels compared to young people and ethnic minority groups tend to have higher stress levels compared to white people [44]. Additionally, marital status and income have been linked to stress levels, where studies report that women who were married and/or had a lower income tended to have higher stress levels than those who were single and/or had a high income [45,46]. We hypothesized that the country context/culture could also affect stress. For example, some studies have pointed out that the cultural context affects the types of stressors, the appraisal, the coping strategies, and the mechanisms of coping with stress [47]. Physical activity has been shown to lower the levels of stress and is related to better stress-coping abilities [48]. Conversely, adiposity (greater BMI) has been found to be positively associated with stress levels, where research has shown that overweight and obese people tended to have poorer metabolic health and increased social problems (like peer/workplace discrimination) along with elevated physiological stress response compared to normal weight people [40,49]. With regard to stress and diet, a recent systematic review and meta-analysis by Khaled et al. (2020) [31] found that stress was associated with poorer diet quality and patterns. This was evidenced by the increased intake of unhealthy food such as high fat/high sugar foods among women of childbearing age (18–49 years old) [31]. Other studies have also reported an association between stress and poor dietary quality and patterns in childbearing-aged women [50,51,52]. Higher age, income, and physical activity levels along with marital status (being married) have been linked to a higher diet quality and healthy patterns [53,54,55,56], whereas ethnicity (e.g., African American) and greater adiposity were linked to poorer dietary quality and patterns [57,58]. Based on this thorough evidence, the hypothesised model is summarized in Figure 2.

### 2.4. Statistical Analysis

Statistical analysis was carried out using IBM SPSS statistics version 28 (Chicago, IL, USA) and AMOS version 28 Graphics (SPSS Inc., Chicago, IL, USA). This study used Structural Equation Modelling (SEM), a comparatively recent method for assessing conceptual models by quantifying the links and interactions among a network of variables [30,59].

The normality of the whole dataset was assessed by computing normality plots and deriving descriptive measures of skewness/kurtosis, and proper transformations were applied to enhance fit normality [60].

The adherence to Mediterranean diet (MD) (a priori dietary method) was derived from nine of the food groups (grams per day) that were calculated from the EPIC FFQ: fruits and nuts, vegetables, legumes, cereals, fish and seafood, alcoholic beverages, meat and meat products, milk and dairy products, and the ratio of unsaturated to saturated fats. Participants whose intake of fruits and nuts, vegetables, legumes, cereals, and seafood, and the unsaturated to saturated fat ratio were below the median were assigned a score of 0, while those meeting or exceeding the median received a score of 1. Conversely, participants whose consumption of meat and meat products, as well as milk and dairy products, was below the median were assigned a score of 1, whereas those whose intake was at or above the median were assigned a score of 0 [10]. A higher score resembles higher diet quality and healthy adherence to the Mediterranean dietary pattern. For the a posteriori dietary method, factor analysis was conducted for the food groups derived from the EPIC FFQ to reveal the latent dietary patterns. The results of the Kaiser–Meyer–Olkin (KMO) test and Bartlett’s test of sphericity indicated a large KMO of 0.828 (>0.5) and a very significant Bartlett’s test of sphericity (*p* < 0.001) denoting the appropriateness of conducting factor analysis. A varimax rotation was performed to calculate factor loadings and a scree plot was drawn to show the number of factors (dietary patterns) with an eigenvalue greater than 1 to be retained. The a priori dietary quality and a posteriori latent dietary patterns of the total sample were included in the last stage of statistical analysis (in the hypothetical model which was tested through the Structural Equation Modelling technique discussed below).

After that, the data were prepared for performing Structural Equation Modelling (SEM). There are five key steps to SEM: (1) identify the research problem, (2) identify the model, (3) estimate the model, (4) determine the model’s goodness of fit, and (5) respecify the model if needed [59].

Next, the SEM model was estimated and tested using data from two countries through assessing the model-fit, estimating the path coefficients (hypothesis testing), and estimating the squared multiple correlations (R^2^) [61]. To assess the most suitable fitting model for the study’s data, the following fit indices were computed: comparative fit index (CFI) > 0.90, chi-square test (χ^2^)/degrees of freedom (df) ratio < 5, standardized root mean square residual (SRMR) <0.08, parsimony normed fit index (PNFI) > 0.5, root mean square error of approximation (RMSEA) ≤ 0.08, parsimony comparative fit index (PCFI) > 0.5, goodness of fit index (GFI) > 0.9, and adjusted goodness of fit index (AGFI) > 0.8 [62,63,64]. Model re-specification was carried out to enhance the goodness of fit in the fifth step of the analysis and bootstrapping was applied to test the significance of indirect effects [65].

## 3. Results

Overall, participants from UK had a higher average age, BMI, and physical activity level compared to participants from Lebanon (Table 1). On the other hand, the stress level, represented by average stress score, was found to be higher among participants from Lebanon compared to those from UK. Most participants from the two countries were non-smokers, unmarried, and had income below average. However, 73% of participants from UK were White, while 100% of participants from Lebanon were Arab.

### 3.1. A-Priori Diet Quality and General Characteristics of Participants across the Diet Quality Categories

Table 2 presents the descriptive data on categorical (ethnicity, marital status, smoking status, and income) and numerical variables (stress scores, physical activity, BMI, and age) across the diet quality categories (low, medium, and high adherence to MD). The adherence to Mediterranean diet of the total sample appeared to be moderate where 58% of participants had a medium adherence to MD, 29% had low adherence, and 13% had high adherence.

### 3.2. A-Posteriori Dietary Patterns

Factor analysis revealed four dietary patterns among participants from the two countries, and Table 3 displays the factor loadings of the food groups within each dietary pattern. The first dietary pattern (DP 1) had highest factor loadings for alcohol, cereals, fats and oils, and sugar and snacks food groups. Dietary pattern 2 (DP 2) had highest factor loadings for vegetables, legumes, soups and sauces whereas dietary pattern 3 (DP 3) had highest factor loadings for eggs, fish and seafood, meats, and potatoes food groups. The last dietary pattern (DP 4) had highest factor loadings for fruits, nuts and seeds, milk and dairy products, and beverages (non-alcoholic).

### 3.3. Structural Equation Modelling (SEM)

The hypothesized model (SEM) revealed the significant direct and indirect pathways between sociodemographic characteristics, physical activity, BMI, stress, and dietary quality and patterns of participants (Table 4). The model fit indices showed good fit of the model where CMIN/DF = 3.215, standardized RMR = 0.0594, RMSEA = 0.067, CFI = 0.896, PNFI = 0.608, PCFI = 0.645, AGFI = 0.911, and GFI = 0.942. In addition, a path analysis diagram was generated (Figure 3) to show the standardized estimates of the total effects between variables. Significant effects are demonstrated by the red arrows indicating that the effect has a *p* value < 0.05.

Table 4 and Figure 3 revealed that stress was directly and negatively associated with the adherence to MD (B = −0.115, *p* = 0.007), but not with any of the a posteriori dietary patterns.

The country in which participants lived was directly associated with their stress levels (B = 0.151, *p* = 0.007) and adherence to MD (B = 0.475, *p* < 0.001), DP 2 (B = 0.225, *p* < 0.001), DP 3 (B = 0.141, *p* = 0.013), and DP 4 (B = −0.245, *p* < 0.001). Being from Lebanon was associated with having significantly higher stress levels and higher adherence to the Mediterranean diet, with significantly greater intakes of food groups such as vegetables, legumes, and soups and sauces and a lower intake of eggs, fish, meats, potatoes, fruits, nuts and seeds, dairy products, and non-alcoholic beverages.

Having a higher income was associated with a higher intake of vegetables, legumes, and soups and sauces food groups but an overall lower adherence to Mediterranean diet. This is shown in Table 4, where income was found to be positively associated with DP 2 (B = 0.133, *p* = 0.003) but negatively associated with the adherence to MD (B = −0.084, *p* = 0.04).

Marital status was positively associated only with DP 3 (B = 0.114, *p* = 0.026), and age was positively associated with MD score (B = 0.12, *p* = 0.031) and DP 4 (B = 0.183, *p* = 0.001). In other words, participants who were unmarried had a significantly greater consumption of eggs, fish, meats, and potatoes, and those who were older in age had significantly higher adherence to Mediterranean diet and intake of fruits, nuts and seeds, dairy products, and non-alcoholic beverages compared to younger participants.

Additionally, ethnicity was negatively associated with adherence to MD (B = −0.103, *p* = 0.029) and DP 4 (B = −0.111, *p* = 0.022) but positively associated with DP 3 (B = 0.168, *p* = 0.029). Participants who were Arabs had a lower intake of fruits, nuts and seeds, dairy products, and non-alcoholic beverages and lower adherence to MD but a higher intake of eggs, fish and seafood, meats, and potatoes compared to other ethnic groups.

BMI was found to be directly and positively associated with DP 3 and DP 4 (B = 0.172, *p* < 0.001 and B = 0.266, *p* < 0.001, respectively), where participants with greater BMI tended to consume more eggs, fish, potatoes, fruits, meats, nuts and seeds, dairy products, and non-alcoholic beverages.

The indirect effect revealed by modelling SEM indicated that among all exogenous variables, only country was found to be indirectly associated with dietary quality and patterns through the mediatory effect of stress (Table 4). The country of participants was found to have an indirect effect of MD score (B = −0.017, *p* = 0.005), DP 1 (B = 0.01, *p* = 0.011), DP 3 (B = 0.01, *p* = 0.016), and DP 4 (B = −0.013, *p* = 0.031) via the mediatory effect of stress. Greater stress levels associated with the country context in the present study played an important role in contributing to dietary quality and patterns. Stress has been found to negatively impact the adherence to Mediterranean diet (diet quality), in addition to dietary patterns that consist of fruits, nuts and seeds, dairy products, and non-alcoholic beverages. Moreover, stress contributed to a higher consumption of dietary patterns that included fats and oils, sugar and snacks, cereals, and alcohol, in addition to eggs, fish, meats, potatoes, when participants were from Lebanon compared to UK.

No association was found between any of the other sociodemographic characteristics and dietary quality and patterns of participants, neither between physical activity nor BMI.

## 4. Discussion

Contrary to the popular thinking and expectation that countries in the Mediterranean basin have healthier diets compared to western countries, the findings of this paper indicated poorer diet quality and patterns in Lebanon due to the high levels of stress. Country comparison using standard methodology across different countries has been recognised as a crucial approach in nutritional epidemiology [13]. Such research ensures a sufficient number of participants, reduces selection bias, fills the gap in the literature, and provides a clearer insight into the problem [13]. The present study adds to the body of knowledge by advancing the understanding of mechanisms through which dietary quality and patterns of women of childbearing age are affected. Our approach has enabled a thorough examination of the pathways linking sociodemographic characteristics, physical activity, and adiposity to dietary quality and patterns through the mediating effect of stress.

In the total sample of women of childbearing age from the UK and Lebanon, we found that stress was negatively associated with the adherence to MD (Table 4). This is in line with the findings of other studies, among women of childbearing age, which report negative associations between stress levels and dietary quality indices (e.g., Alternate Healthy Eating Index 2010 [50] and Dietary Quality Index—Pregnancy [51,52]). In contrast, some studies found no association between stress levels and dietary quality among women of childbearing age [66,67]. However, these studies had small sample sizes, used different methodology (e.g., 24 h recalls for dietary assessment), and were conducted in a single country (e.g., Egypt). The present study also found that the sociodemographic characteristics of participants influenced dietary quality and patterns where higher income was found to be linked with a higher intake of vegetables, legumes, and soups and sauces. Additionally, being unmarried was found to be linked with greater consumption of eggs, fish, meats, and potatoes, and age was found to be linked with higher adherence to Mediterranean diet and intake of fruits, nuts and seeds, dairy products, and non-alcoholic beverages. Ethnicity (being Arab compared to other ethnicities within the study’s sample) was found to be linked with a lower adherence to MD and intake of fruits, nuts and seeds, dairy products, and non-alcoholic beverages, but with a higher intake of eggs, fish and seafood, meats, and potatoes. These findings have also been reported in the literature where higher age and income have been linked with a higher diet quality and healthy patterns [53,54,55,56]. In addition, ethnicity (e.g., African American) and being unmarried were linked to poorer dietary quality and patterns [57,58]. The present study found that BMI was positively associated with greater consumption of eggs, fish, potatoes, fruits, meats, nuts and seeds, dairy products, and non-alcoholic beverages. This confirms previous evidence that greater adiposity (BMI) is linked to lower intake of fruits, vegetables, and legumes, and higher intake of high sugary foods, snacks, and high fat foods [57,58].

While many factors were found to have a direct effect on dietary quality and patterns, only country was found to be indirectly associated with dietary quality and patterns through the mediatory effect of stress. Participants from Lebanon were found to have higher stress levels compared to participants from the UK, and this contributed to a lower adherence to Mediterranean diet. Stress also contributed to lower consumption of eggs, fish, meats, potatoes, fruits, nuts and seeds, dairy products, and non-alcoholic beverages, and higher consumption of fats and oils, sugar and snacks, cereals, and alcohol. These findings indicate that lower stress levels are associated with nutritionally dense diets, whilst higher stress levels contribute to less nutritionally dense and diverse diets. These are intriguing observations and have not been previously reported in the literature.

The high rate of stress can be explained by the fact that Lebanon, a low/middle-income country that hosts around two million refugees, has passed through various stressful events during the past decade. These include, but are not limited to, the COVID-19 pandemic and the explosion of the port of Beirut in August 2020, which was the world’s biggest non-nuclear explosion of the 21st century [68]. In addition, Lebanon has been experiencing economic crisis and collapse since 2019. Due to the economic crisis, Lebanon has not been able to import medicines and medical equipment, and many people have lost their jobs. These events have contributed to the elevated levels of stress among the population, who remain untreated [69]. A survey collected in 2022 indicated that among 903 adults in Lebanon, 83% reported feeling sad and stressed at almost all times of the day and 11.5% reported having suicidal ideation [70]. Moreover, a recent study has found that Lebanese people had decreased intakes of fruits, vegetables, and water and increased intakes of fats and sugars, which led to increased weight gain [71].

## 5. Conclusions

This paper represents the first study to examine the association between stress, dietary quality and patterns, and sociodemographic characteristics, along with adiposity and physical activity among women of childbearing age, using Structural Equation Modelling. To the best of our knowledge, the mediatory role of stress in the relation between sociodemographic characteristics, adiposity, and physical activity with dietary quality and patterns, specifically among childbearing aged women, has not been tested before. Moreover, analysing data from two different countries/continents has filled the gap reported in the literature, where many papers have previously recommended new studies to include a diverse population when assessing the association between diet and other predictors [22]. Confidence in the study results is acquired by the large sample size, consistent methodology across the two countries (UK and Lebanon) that avoided bias, and the application of a powerful statistical technique (Structural Equation Modelling). By simultaneously modelling mediating pathways, the present study’s analyses were able to derive the total, direct, and indirect effects of the different predictors of dietary quality and patterns, advancing prior research that has examined the association between diet and specific variables/predictors in isolation. The sample size of the present study was relatively big (*n* = 493), and this is highly desirable since the Structural Equation Modelling statistical technique is dependent on sample size [30,62]. Additionally, studies in UK and Lebanon used validated and standardized tools to obtain the data (such as the EPIC FFQ for assessing dietary intake, PSS for measuring stress levels, and IPAQ for measuring physical activity), which is also an important strength. Furthermore, the present study applied two approaches of dietary analyses: the a priori (hypothesis-driven) and the a posteriori (data-driven), which describe the overall dietary quality and patterns of participants more precisely. This is an improvement over previous studies, which have been limited because they only measured the intake of individual nutrients (e.g., selenium) or foods/food groups (e.g., fat, sugar, fruits). It is worth acknowledging that the present study has some limitations. The participants were from university settings in both UK and Lebanon, which makes it difficult to generalise the results to the whole population of women of childbearing age. Furthermore, since the data collected in the surveys were self-reported, potential biases from misreporting may have occurred.

We used Structural Equation Modelling to investigate the association between stress and diet among women of childbearing age allowing for country comparison. The findings suggest that the dietary quality and patterns of childbearing aged women are affected by the country context, and this can be explained by stress levels. However, further research is needed to explore the impact of dietary recommendations and their cultural adaption to diverse populations [72]. At a clinical level, these findings will inform the implementation of interventions (such as stress coping strategies) intended to lower stress levels, and hence improve the diet of the population. Identifying whether tailoring interventions on the basis of lower stress levels leads to improved diet quality and patterns remains to be examined in well-designed randomised controlled trials.

## Figures and Tables

**Figure 1 nutrients-16-03354-f001:**
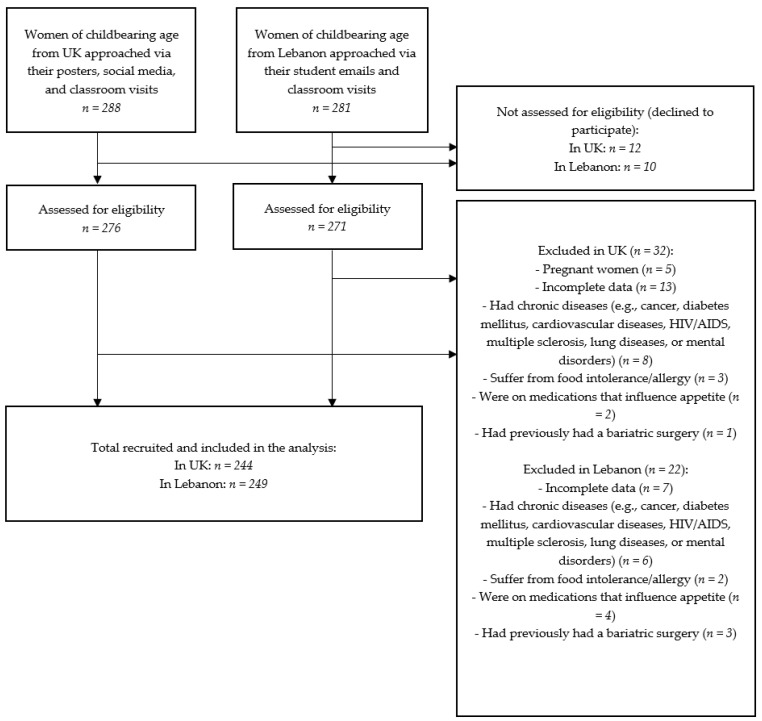
Flow diagram of participant enrolment in the studies.

**Figure 2 nutrients-16-03354-f002:**
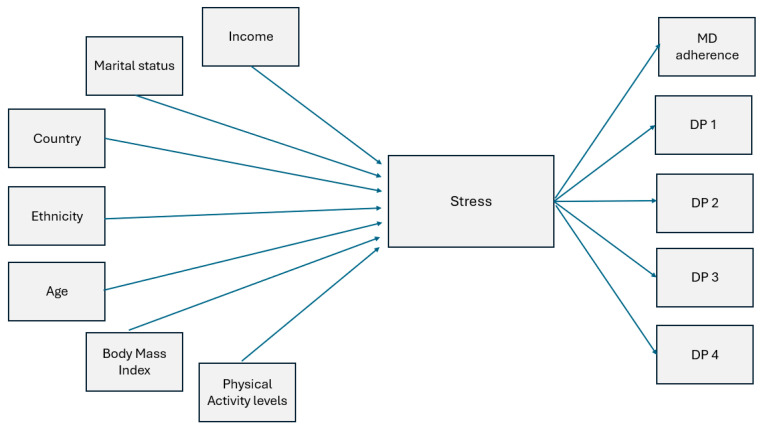
General hypothesised model showing stress as a mediating variable relating sociodemographic characteristics, adiposity, physical activity, and country context to dietary quality (Mediterranean diet (MD) adherence) and patterns (DPs derived from factor analysis). Abbreviations: MD, Mediterranean diet; DP, dietary pattern.

**Figure 3 nutrients-16-03354-f003:**
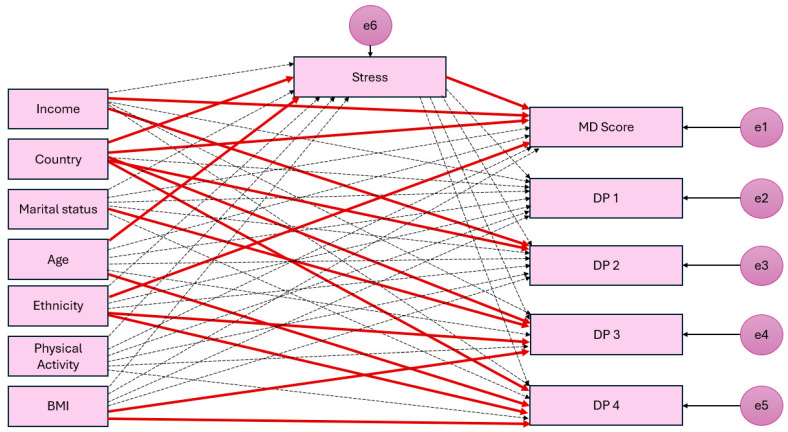
Simplified path analysis diagram highlighting the association of sociodemographic characteristics, physical activity, and BMI with dietary quality and patterns. Red arrows are effects with *p*-value < 0.05. Dotted lines show non-significant effects. The standardized estimates are shown in Table 4, and the detailed diagram is available from first author (K.K.). Abbreviations: Mets, metabolic equivalents of tasks (hours per week); BMI, Body Mass Index; MD, Mediterranean diet; DP, dietary pattern.

**Table 1 nutrients-16-03354-t001:** Characteristics of study participants by country.

Variables		Country	
	UK (Europe)	Lebanon (Mediterranean Region)	*p*-Value
Age (years) *	24.0 (21.0–32.0)	19 (18–21)	0.016
Stress score *	29 (22.0–33.0)	31.6 (17.1–36.2)	0.285
BMI (kg/m^2^) *	23.7 (20.9–27.9)	22.5 (20.1–25.6)	0.579
Physical activity (METs-h/wk) *	1429 (464.3–2824.5)	990 (346.5–2170.5)	<0.01
Income per year (N (%))			-
Below average	119 (49)	160 (64.3)	
Average	99 (40)	52 (20.9)	
Above average	26 (11)	37 (14.9)	
Marital status (N (%))			-
Married	43 (18)	6 (2.4)	
Unmarried	201 (82)	243 (97.6)	
Smoking (N (%))			-
Smoker	56 (23)	64 (25.7)	
Non-smoker	188 (77)	183 (73.5)	
Ethnicity (N (%))			-
White	177 (73)	0 (0)	
Black	15 (6)	0 (0)	
Asian	35 (14)	0 (0)	
Arab or other ethnic group	17 (7)	249 (100)	

METs-h/wk: Metabolic equivalents of task-hours per week, BMI: Body Mass Index. * Data represent median (interquartile range).

**Table 2 nutrients-16-03354-t002:** Descriptive characteristics of the total sample of childbearing-aged women from UK and Lebanon (*n* = 493).

Variables		Low Adherence (MD Score: 0–3)	Medium Adherence (MD Score: 4–6)	High Adherence (MD Score: 7–9)	Total
Total *		143 (29.0)	288 (58.4)	62 (12.6)	493 (100.0)
N (%)					
Ethnicity *	Arab	36 (14.5)	167 (67.1)	46 (18.4)	249 (50.5)
	Asian	18 (54.5)	14 (42.5)	1 (3.0)	33 (6.7)
	Black	8 (53.3)	6 (40.0)	1 (6.7)	15 (3.0)
	White	81 (41.3)	101 (51.5)	14 (7.2)	196 (39.8)
Marital status *	Married	16 (32.7)	27 (55.1)	6 (12.2)	49 (9.9)
	Unmarried	127 (28.6)	261 (58.8)	56 (12.6)	444 (90.1)
Smoking *	Non-smokers	107 (28.4)	226 (59.9)	44 (11.7)	377 (76.5)
	Smokers	36 (31.0)	62 (53.4)	18 (15.6)	116 (23.5)
Income *	Average Income	65 (42.5)	75 (49.0)	13 (8.5)	153 (31.0)
	High Income	13 (20.6)	43 (68.3)	7 (11.1)	63 (12.8)
	Low Income	65 (23.5)	170 (61.4)	42 (15.1)	277 (56.2)
Median (interquartile range)					
Age ^		21 (19–27)	21 (18–25)	19 (18–24)	21 (19–25)
Stress score ^		31 (26.5–34)	30 (24–35)	31 (27–34)	30 (25.535)
Physical activity METs ^		1039 (470–1269.7)	1200 (396–2717.2)	1293 (280.5–2772)	1217 (381–2579)
BMI ^		24.34 (20.9–28.7)	22.5 (20.3–26)	22.94 (20.4–25.7)	22.84 (20.4–26.7)

* Data for categorical variables presented as N (%). ^ Data for numerical variables presented as median (interquartile range). METs: metabolic equivalents of tasks (hours per week).

**Table 3 nutrients-16-03354-t003:** Varimax-rotated factor loadings of the fifteen food groups on the four factors (dietary patterns).

	Factors (Dietary Patterns)			
	1	2	3	4
Alcohol (grams per day)	0.150			
Cereals (grams per day)	0.812			
Eggs (grams per day)			0.469	
Fats and Oils (grams per day)	0.840			
Fish and Sea food (grams per day)			0.676	
Fruits (grams per day)				0.621
Meats (grams per day)			0.887	
Milk and dairy products (grams per day)				0.349
Non-alcohol beverages (grams per day)				0.309
Nuts and seeds (grams per day)				0.424
Potatoes (grams per day)			0.379	
Soups and sauces (grams per day)		0.584		
Sugar and snacks (grams per day)	0.627			
Vegetables (grams per day)		0.615		
Legumes (grams per day)		0.919		

**Table 4 nutrients-16-03354-t004:** Statistically significant pathways (direct and indirect) of the association between sociodemographic characteristics and physical activity and BMI with dietary quality and patterns among women of childbearing age using Structural Equation Modelling.

Model Path	Standardised Estimate	SE	*p*
Direct Effects			
Direct Impact on MD score			
Stress → MD score	−0.115	0.01	0.007
Income → MD score	−0.084	0.108	0.04
Country → MD score	0.475	0.193	<0.001
Age → MD score	0.120	0.14	0.031
Ethnicity → MD score	−0.103	0.133	0.029
Country → stress	0.151	0.859	0.007
Direct Impact on DP 2 (DP 2 food groups: vegetables, legumes, soups, and sauces)			
Income → DP 2	0.133	0.06	0.003
Country → DP 2	0.225	0.107	<0.001
Direct Impact on DP 3 (DP 3 food groups: eggs, fish and seafood, meats, and potatoes)			
Country → DP 3	−0.141	0.108	0.013
Ethnicity → DP 3	0.168	0.075	0.029
Marital status → DP 3	0.114	0.163	0.026
BMI → DP 3	0.172	0.0	<0.001
Direct Impact on DP 4 (DP 4 food groups: fruits, nuts and seeds, dairy products, and non-alcoholic beverages)			
Country → DP 4	0.245	0.093	<0.001
Age → DP 4	0.183	0.007	0.001
Ethnicity → DP 4	−0.111	0.064	0.022
BMI → DP 4	0.266	0.0	<0.001
Indirect Effects via Stress			
Country → MD score	−0.017	0.010	0.005
Country → DP 1	0.01	0.006	0.011
Country → DP 3	0.01	0.006	0.016
Country → DP 4	−0.013	0.008	0.031
Residual Covariance			
Ethnicity and Age	−1.026	0.204	<0.001
Ethnicity and Country	0.152	0.016	<0.001
Age and Country	−1.799	0.176	<0.001
Country and Marital	0.038	0.007	<0.001
Marital and Income	−0.025	0.01	0.01
Age and Marital	−1.083	0.105	<0.001
Age and Income	1.013	0.226	<0.001

Abbreviations: Mets, metabolic equivalents of tasks (hours per week); BMI, Body Mass Index; MD, Mediterranean diet; DP, dietary pattern.

## Data Availability

Bournemouth University aims to make our research data as openly accessible as possible. Data will be registered and discoverable via BORDaR (https://bordar.bournemouth.ac.uk), BU’s research data repository, and linked (where applicable) to any associated research outputs via BURO, BU’s institutional repository.

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
