# Peer review of "A Structural Equation Modelling Approach to Examine the Mediating Effect of Stress on Diet in Culturally Diverse Women of Childbearing Age"

_nutrients, 2024, doi:10.3390/nu16193354_

Round 1

Reviewer 1 Report

Comments and Suggestions for Authors

This is an important study that employs sophisticated methods to examine the effects of perceived stress on diet quality.  Please see the comments meant to strengthen the paper.

Introduction 

Add literature describing diet quality with a-priori and a-posteriori approaches of the target population. After reading the introduction I am not clear on the level of adherence to dietary patterns in general, which dietary patterns they or Mediterranean in the target population, or the success with diet intervention trials

Why did you conduct a-priori and a-posteriori approaches for Mediterranean diet scores? Are the county context the audience should be aware of? Please explain.

Methods

I would suggest adding section focused on dietary assessment with correspond subsections. All dietary assessment measures need to be described including scoring approaches for a-priori methods. Explicitly state a higher score means healthy adherence to Mediterranean dietary patterns. 

Are there significant clinal cutoffs regarding the diet score the audience should be aware of? It would be helpful for the public health relevance and integration of your finding

Clarify race and race vs ethnicity in your sample demographic. For example White is not an ethnicity it is a race. Similar to Black. 

At the Blacks in your sample African American? What is their ethnicity? The terms are not necessarily interchangeable.

Results

Clarify of race and ethnicity in Table information

Add scale for range and interpretive informative to application measures like  perceived stress, Med diet score

Discussion

I appreciate the author's efforts to discuss contextual factors regarding the country in their study. 

With clarified information on race vs ethnicity, discuss disparities as we as health equity implications for your research findings

Comments on the Quality of English Language

The paper could benefit from revisions mentioned and then additional proofread reading. 

Author Response

We are grateful to the reviewer for their detailed feedback that has greatly improved the manuscript. The changes made are detailed point by point below and highlighted in the manuscript in yellow.

Comment from reviewer 1

Response

This is an important study that employs sophisticated methods to examine the effects of perceived stress on diet quality.

Thank you.

Introduction

Add literature describing diet quality with a-priori and a-posteriori approaches of the target population. After reading the introduction I am not clear on the level of adherence to dietary patterns in general, which dietary patterns they or Mediterranean in the target population, or the success with diet intervention trials

The literature requested has been added to the introduction and the references have been amended accordingly. Please see lines 48-64.

Why did you conduct a-priori and a-posteriori approaches for Mediterranean diet scores? Are the county context the audience should be aware of? Please explain.

We took an a-priori and a-posteriori approach to overcome the limitations of a single dietary scale. The a-priori approach was based on assessing the adherence to Mediterranean Diet, and the a-posteriori approach was based on performing factor analysis to derive the latent dietary patterns of participants. This has now been addressed in the paper (lines 48-64).

As nutritionists, we have critiqued the dietary assessment tools used in previous studies in the background of the paper. Based on that, and because dietary assessment is an important aspect in such studies, we have used a validated food frequency questionnaire that has over 140 food groups instead of single item dietary tools. The European Prospective into Cancer food frequency questionnaire used in our study is deemed to be the gold standard in nutrition epidemiology and has been widely used. By assessing diet in two ways, a-priori and a-posteriori, we have a greater understanding about the overall diet quality and patterns of our participants in both countries.

Methods

I would suggest adding section focused on dietary assessment with correspond subsections. All dietary assessment measures need to be described including scoring approaches for a-priori methods. Explicitly state a higher score means healthy adherence to Mediterranean dietary patterns. 

This has been addressed in text as recommended. Please see lines 212-227.

Are there significant clinical cutoffs regarding the diet score the audience should be aware of? It would be helpful for the public health relevance and integration of your finding

This has been added as suggested. Please see lines 212-227.

Results

Clarify race and race vs ethnicity in your sample demographic. For example White is not an ethnicity it is a race. Similar to Black. 

This has been amended to reflect terminology used in the citation (line 177).

In table 1 - We have used the official UK ethnicity classifications. Please see:

https://www.ethnicity-facts-figures.service.gov.uk/style-guide/ethnic-groups/

Add scale for range and interpretive informative to application measures like perceived stress, Med diet score

This information has been added as suggested. Please see lines 151-155.

Discussion

I appreciate the author's efforts to discuss contextual factors regarding the country in their study. 

Thank you.

With clarified information on race vs ethnicity, discuss disparities as we as health equity implications for your research findings

We have clarified the use of terminology regarding ethnicity (see above).

We have added a comment and citation in the discussion to address the point on health equity (lines 443-444).

Reviewer 2 Report

Comments and Suggestions for Authors

An interesting study that explores, through a theoretically supported model, whether stress influences nutritional patterns in two cohorts, UK and Lebanon. However, some comments lead me to read this work with caution.

Firtly, the context of the population needs greater definition, since stress is led by the number of previous abortions, multiparity, type of birth, family nucleus, perceived social supports, etc. which have not been considered in this work.

Comments:

- The introduction should be summarized and focused. For example, lines 59 to 61 the idea is repeated in the study by Barrington et al [12], lines 63-64.

- In lines 70-71 it is mentioned that one of the limitations of the studies is using a single dietary scale, however, the authors also only use a single stress perception scale, which, although widely used, explores one dimension.

- Lines 95-97, samples from the same continent have sociodemographic or stress similarities, even within the same country, social and nutritional variability is very great.

- "Study population" section, please present a flow chart with the enrollment, when they were excluded, why they were excluded in each case, etc. Also, indicate that the study follows the indications of STROBE? (https://www.strobe-statement.org/). 

- The variables have to be described deeply, how were the indices extracted? How is the Mediterranean diet adherence calculated? Why the categories that are later described in the results? How confident is the PSS, how is it scored? How were the METs extracted? Was adiposity self-reported? this may be a bias. On the other hand, is adiposity considering weight and height? this cannot be estimated as adiposity.

- Lines 159-181, this should be clear in the intro. If the authors wanted to report it here, it should be extremely concise.

- Table 1 should give the univariate contrasts between the two cohorts, with the exact P-value. Likewise in table 2.

- Table 3, alcohol has a very low correlation, should not it have escaped factor 1? Did you use some kind of matrix rotation? What variability explains factor 1?

- Figure 2. It is not at all understandable, I suggest leaving the significant red lines, and only the interactions between the dependent variables, stress and nutritional patterns. On the other hand, this effect could be led by the country of the cohort, and not by stress.

- Lines 316-318, I think they are independent indications, a country can have a healthier pattern and its population can be stressed and this can make the individual pattern less healthy. 

Minor comments:

- Rephrase the introduction, lines 50 and 53 begin with "understanding"

- JISC: putting the acorn name has proven to be a valid method of collecting information?

Comments on the Quality of English Language

Paraphrase and focus some paragraphs.

Author Response

We are grateful to the reviewer for their detailed feedback that has greatly improved the manuscript. The changes made are detailed point by point below and highlighted in the manuscript in yellow.

Comment from reviewer 2

Response

An interesting study that explores, through a theoretically supported model, whether stress influences nutritional patterns in two cohorts, UK and Lebanon.

Thank you.

Firstly, the context of the population needs greater definition, since stress is led by the number of previous abortions, multiparity, type of birth, family nucleus, perceived social supports, etc. which have not been considered in this work.

We agree that there are multiple factors that can influence stress. We have tried to limit these by excluding participants with chronic diseases, food intolerance/allergy, medications that impact appetite, or have had bariatric surgery (section 2.1 – lines 125-132). In addition, we excluded participants who were pregnant or breastfeeding. 

The introduction should be summarized and focused. For example, lines 59 to 61 the idea is repeated in the study by Barrington et al [12], lines 63-64.

This has been summarised as suggested (see lines 66-69).

In lines 70-71 it is mentioned that one of the limitations of the studies is using a single dietary scale, however, the authors also only use a single stress perception scale, which, although widely used, explores one dimension.

We took an a-priori and a-posteriori approach to overcome the limitations of a single dietary scale. The a-priori approach was based on assessing the adherence to Mediterranean Diet, and the a-posteriori approach was based on performing factor analysis to derive the latent dietary patterns of participants. This has now been addressed in the paper (see lines 48-64).

Lines 95-97, samples from the same continent have sociodemographic or stress similarities, even within the same country, social and nutritional variability is very great.

This has been rewritten to clarify the reference to the specific study (see lines 103-105).

"Study population" section, please present a flow chart with the enrollment, when they were excluded, why they were excluded in each case, etc. Also, indicate that the study follows the indications of STROBE? (https://www.strobe-statement.org/). 

This has been added as suggested and in-text references of the paper have been amended accordingly (see line 132-133).

We have added a flowchart for the enrolment as a supplementary material at the end of the paper (see line 651).

The variables have to be described deeply, how were the indices extracted? How is the Mediterranean diet adherence calculated? Why the categories that are later described in the results? How confident is the PSS, how is it scored? How were the METs extracted? Was adiposity self-reported? this may be a bias. On the other hand, is adiposity considering weight and height? this cannot be estimated as adiposity.

This has been addressed in text as recommended (see lines 209-223).

Additional  information on variables has been added (see lines 148 -152).

Lines 159-181, this should be clear in the intro. If the authors wanted to report it here, it should be extremely concise.

The rich description included in this section is important in explaining the assumptions that underpin the SEM model.

We are aware that lines 159-181 present a discussion of literature, and usually literature is included in the introduction of a paper. However, using Structural Equation Modelling requires a clear description of the evidence base behind building the model.

Therefore, we believe that this section best fits here as it is part of the steps used in Structural Equation Modelling and hence would seem unrelated if it was included in the introduction and hard to refer to it when the reader reaches the methods describing the steps of Structural Equation Modelling.

Table 1 should give the univariate contrasts between the two cohorts, with the exact P-value. Likewise in table 2

This information has been added as suggested. Please see line 254.

Table 3, alcohol has a very low correlation, should not it have escaped factor 1? Did you use some kind of matrix rotation? What variability explains factor 1?

As part of the factor analysis and Varimax rotation, the numbers presented in the table describe factor loadings for the food group that they belong to. So, the highest factor loading of alcohol was for factor one. This applies to all other food groups where the numbers presented in the table represent the factor loadings for factors (dietary patterns) where they scored highest.

Figure 2. It is not at all understandable, I suggest leaving the significant red lines, and only the interactions between the dependent variables, stress and nutritional patterns. On the other hand, this effect could be led by the country of the cohort, and not by stress.

For clarity, we have presented the significant findings (demonstrated in red arrows in figure 2) in table 4.

Figure 2 provides the complete model. The path analysis diagram in structural equation modelling needs to show the paths of each variable (significant and insignificant paths). For this reason, we believe that the figure should include all paths that provide necessary information about the model.

Lines 316-318, I think they are independent indications, a country can have a healthier pattern and its population can be stressed and this can make the individual pattern less healthy. 

The authors agree with this point and have mentioned this in the discussion of the paper.

Minor comments:

Rephrase the introduction, lines 50 and 53 begin with "understanding"

This has been rephrased.

JISC: putting the acorn name has proven to be a valid method of collecting information?

This has been reworded.

Comments on the Quality of English Language:

Paraphrase and focus some paragraphs.

Some sections have been rephrased and are more focused now.

Round 2

Reviewer 2 Report

Comments and Suggestions for Authors

Thank you for this review of the manuscript and responses to suggestions.

- The flow chart should appear within the main text rather than as supplementary material.

- On the other hand, figure 2 remains unclear.

Comments on the Quality of English Language

There are difficult to understand sentences and typographical and stylistic errors throughout the manuscript.

Author Response

We are grateful to the reviewers for their detailed feedback that has greatly improved the manuscript. The further changes made are detailed point by point below and highlighted in the manuscript in yellow.

The flow chart should appear within the main text rather than as supplementary material.

The flow chart has now been moved to the main text, and the figure numbers and text have been amended accordingly. Please see lines 151 - 191.

On the other hand, figure 2 remains unclear.

Figure 2 (now Figure 3) has been simplified as requested. We have retained the paths of each variable (significant and insignificant paths) but removed the standardised estimates, which are given in Table 4. We have added a footnote to indicate that the full path analysis diagram can be obtained from the first author.

Comments on the Quality of English Language:

There are difficult to understand sentences and typographical and stylistic errors throughout the manuscript.

The corresponding author is a native English speaker and has thoroughly reviewed the manuscript.

Round 3

Reviewer 2 Report

Comments and Suggestions for Authors

Thank you for this revised version of the manuscript.